 Medicine

# Childhood cancer in Sweden during the COVID-19 pandemic: Temporal patterns in incidence and survival in a nationwide register-based cohort study

Christina-Evmorfia Kampitsi[1☉*], Javier Louro[1☉], Hanna Mogensen[1,2], Friederike Erdmann[3,4], Kleopatra Georgantzi[5], Mats Heyman[5], Päivi Lähteenmäki[5], Anna Nilsson[5], Maria Feychting[1], Giorgio Tettamanti[1,6]

**1** Unit of Epidemiology, Institute of Environmental Medicine, Karolinska Institutet, Stockholm, Sweden, **2** Department of Immunology, Genetics and Pathology, Cancer Precision Medicine, Uppsala University, Uppsala, Sweden, **3** Research Group Aetiology and Inequalities in Childhood Cancer, Division of Childhood Cancer Epidemiology, Institute of Medical Biostatistics, Epidemiology and Informatics (IMBEI), University Medical Center of the Johannes Gutenberg University Mainz, Mainz, Germany, **4** Department of Prevention and Evaluation, Leibniz Institute for Prevention Research and Epidemiology—BIPS, Bremen, Germany, **5** Department of Women's and Children's Health, Karolinska Institutet, Stockholm, Sweden, **6** Department of Molecular Medicine and Surgery, Karolinska Institutet, Stockholm, Sweden

☉ These authors contributed equally to this work.
* christina.evmorfia.kampitsi@ki.se

## Abstract

### Background

The COVID-19 pandemic raised concerns about diagnostic delays and treatment disruptions in childhood cancer, potentially threatening survival. We assessed childhood cancer incidence and survival in Sweden, where only few restrictions were implemented, during the pandemic period.

### Methods and findings

We conducted a nationwide, register-based cohort study including all children and adolescents (0–19 years) with a new cancer diagnosis, defined according to the International Classification of Childhood Cancer, Third Edition (ICCC-3), reported to the Swedish National Cancer Register during 2015–2022 ($N = 3,333$; 2,069 pre-pandemic and 1,264 during the pandemic). We compared quarter-specific age-standardized incidence rates (ASR) per 1,000,000 (overall and by diagnostic group) during 2020–2022 to the 2015–2019 average. Overall survival at 3, 6, and 12 months post-diagnosis was calculated using the Kaplan–Meier estimator, and mortality at the same intervals was assessed with logistic regression, adjusted for age at diagnosis, sex, and maternal education.

Overall incidence rates remained largely stable during the pandemic ($ASR_{2015-2019}$: 179.5; 95% CI [172.9,186.1], $ASR_{2020}$: 174.7; 95% CI [160.5,189.0], $ASR_{2021}$: 176.5;

**Data availability statement:** Swedish laws and regulations do not allow sharing of personal sensitive data, which can only be made available for researchers who fulfill legal requirements for access to such data. Eligible individuals can apply for the data from the National Board of Health and Welfare in Sweden (https://bestalladata.socialstyrelsen.se/) and from Statistics Sweden (https://www.scb.se/vara-tjanster/bestall-data-och-statistik/). The data was accessed and verified by Javier Louro and Giorgio Tettamanti. The analysis code is available from Github (https://github.com/jlouroae/R_Code_IAS) and archived in Zenodo (https://doi.org/10.5281/zenodo.17720518).

**Funding:** This project was supported by grants from the Swedish Childhood Cancer Fund (https://www.barncancerfonden.se/en; awarded to GT; grant nr PR2021-0056) and the Swedish Research Council (https://www.vr.se/; awarded to GT; grant nr 2022-06312). The funding sources were not involved in the conceptualization, design, content, or preparation of the manuscript, or the decision to submit for publication.

**Competing interests:** The authors have declared that no competing interests exist.

**Abbreviations:** ALL, acute lymphoblastic leukemia; AML, acute myeloid leukemia; aOR, adjusted odds ratios; ASR, age-standardized incidence rates; CIs, confidence intervals; CNS, central nervous system; ICCC-3, International Classification of Childhood Cancer, Third Edition.

95% CI [161.6,191.5], $ASR_{2022}$: 181.2; 95% CI [166.3,196.2]), but diagnostic groups showed differing tendencies. Acute lymphoblastic leukemia (ALL) declined from February 2020, followed by a rebound in 2021. Acute myeloid leukemia (AML) declined throughout 2020–2022 without evidence of a rebound. Lymphomas declined in mid-2020 before returning to pre-pandemic levels. Central nervous system (CNS) tumors transiently increased in 2020. Overall 1-year survival was 94.8% (95% CI [93.9,95.8]) in the pre-pandemic period and 95.9% (95% CI [94.8,97.0]) during the pandemic. No increase in 6-month or 1-year mortality was observed; if anything, point estimates suggested lower 1-year mortality for ALL (aOR = 0.37; 95% CI [0.08,1.21]; $p=0.14$) and CNS tumors (aOR = 0.61; 95% CI [0.31,1.14]; $p=0.13$). The main limitation of this study was statistical uncertainty for certain diagnostic groups due to small case numbers.

## Conclusions

Overall childhood cancer incidence and survival in Sweden showed no major changes during the COVID-19 pandemic. Fluctuations in diagnostic-group-specific incidence may reflect delayed diagnoses or shifts in disease triggers; however, the timing of the ALL decline and the lack of AML rebound challenge this interpretation. Stable or improved survival suggests that any disruptions were not associated with poorer survival up to 1 year after diagnosis.

## Author summary

### Why was this study done?

- Childhood cancer requires timely diagnosis and uninterrupted treatment, and there have been concerns that the COVID-19 pandemic might have disrupted this care.

- Reports from several countries have yielded heterogeneous findings about whether fewer children were diagnosed with cancer during the pandemic; research about survival during this period is particularly scarce.

- Sweden kept schools and most of society open, offering a unique setting to examine patterns in childhood cancer incidence and survival during the pandemic.

### What did the researchers do and find?

- We examined all cancers newly diagnosed in children and adolescents in Sweden from 2015 to 2022, using nationwide health registers (3,333 children and adolescents).

- We compared the number of new diagnoses and survival up to one year after diagnosis during the pandemic years (2020–2022) with the pre-pandemic years (2015–2019).

- Overall, childhood cancer diagnoses did not decrease during the pandemic.

- Some cancers showed transient increases or decreases, but these changes were not accompanied by worse survival outcomes.

## What do these findings mean?

- Childhood cancer care in Sweden likely remained resilient during the pandemic, as no evidence of poorer short-term survival was observed despite concerns about delays or disruptions.

- Temporary changes in diagnoses of specific cancer types may reflect normal fluctuations rather than lasting pandemic-related shifts.

- The findings highlight the value of strong, adaptable healthcare systems in protecting essential services during global crises.

- Longer-term monitoring is needed to understand whether incidence patterns or outcomes change in the post-pandemic era.

## Introduction

Cancer is a leading cause of disease-related mortality among children and adolescents worldwide [1]. Advances in diagnostics, multimodal therapy, and multidisciplinary care have markedly improved survival over past decades [2]. However, maintaining these gains depends on timely diagnosis and uninterrupted treatment. The COVID-19 pandemic placed unprecedented strain on healthcare systems globally, raising concerns about delays in childhood cancer diagnosis, treatment disruptions, and potential consequences for patient outcomes [3].

These concerns are reinforced by reports of substantial declines in cancer diagnoses across regions during the pandemic, suggesting diagnostic delays [4,5]. Although most studies did not assess cancer specifically in children, reductions in pediatric emergency visits and hospital admissions during lockdowns, as observed in Germany and the Netherlands [6,7], imply that childhood cancer diagnoses may also have been affected. Given differences in healthcare infrastructure, welfare systems, pandemic course, and related measures, the extent of disruptions likely varied substantially across countries. Differences in study period further limit direct comparisons. Nevertheless, studies from the United States, Italy, and Norway indicated declines in childhood cancer diagnoses during the pandemic [4,8–11], whereas no such decrease was observed in studies from Greece, France, or Germany [12–14]. Notably, the German study found an unexpected increase in childhood cancer incidence throughout 2020 [14,15]. However, some of these reports included only data from one hospital or a single region rather than nationwide sources. It should also be noted that, where observed, a decrease in diagnoses may not solely reflect diagnostic delays but could also result from reduced circulation of common infections, an effect particularly relevant to acute lymphoblastic leukemia (ALL). The COVID-19 pandemic, with its societal restrictions, could stand as a "natural experiment" to examine the long-standing hypothesis that immune exposures in early life could contribute to ALL pathogenesis [16], a notion previously supported by observations following school closures during the 2003 SARS outbreak in Hong Kong [17].

Beyond timely diagnosis, potential cancer treatment disruptions and their impact on survival are equally concerning. Delays in diagnosis and treatment initiation can lead to more advanced disease at presentation, requiring more intensive therapy and increasing the risk of complications and long-term late effects [18]. While research on childhood cancer survival during the pandemic is limited, modeling studies of adult cancers have projected increased mortality [19].

Sweden presents a unique setting for evaluating the effects of the COVID-19 pandemic on childhood cancer incidence and outcomes. Unlike most European countries, Sweden did not impose strict lockdowns and kept schools and daycare

facilities open, except for a three-month closure for students over 16 years of age starting in March 2020 [20]. Despite this, primary healthcare utilization declined during the first wave of the pandemic, including among children [21]; whether childhood cancer incidence and survival were affected, however, remains unclear. Using national registry data, this study aimed to assess the childhood cancer incidence and survival in Sweden during the COVID-19 pandemic.

## Methods

### Ethics statement

The study was approved by the Swedish Ethical Review Authority (DNR 2023-01187-01). Informed consent was waived, as the study was register-based and all data were pseudonymized. This study is reported as per the Reporting of Studies Conducted using Observational Routinely-Collected Data (RECORD) guideline (S1 Checklist).

### Study population

For this nationwide registry-based study, we identified all children and adolescents aged 0–19 years, newly diagnosed with cancer in Sweden between January 1, 2015, and December 31, 2022. Cancer diagnoses were obtained from the Swedish National Cancer Register, which has recorded all cancer diagnoses since 1958 [22]. Demographic information, including date of birth, sex, and date of death (available through 2023), were retrieved from the Total Population Register.

### Childhood cancer classification

Childhood cancer was defined according to the International Classification of Childhood Cancer, Third Edition (ICCC-3) [23]. In line with this classification, we included non-malignant central nervous system (CNS) tumors and intracranial/intra-spinal germ cell tumors.

Analyses were conducted overall and by diagnostic group: ALL (ICCC-3 group Ia), acute myeloid leukemia (AML, ICCC-3 group Ib), Hodgkin lymphomas (ICCC-3 group IIa), non-Hodgkin lymphomas (ICCC-3 group IIb), CNS tumors (ICCC-3 group III), and non-CNS solid tumors (ICCC-3 groups IV–XII). Leukemias not classified as ALL or AML (ICCC-3 groups Ic-Ie) and lymphomas not classified as Hodgkin or non-Hodgkin (ICCC-3 group IIc) were included only in overall analyses because of small numbers.

### Statistical analysis

To evaluate changes in childhood cancer incidence, we compared the number of new diagnoses per quarter during 2020–2022 with the corresponding quarter averages from 2015–2019. Diagnoses from January 1, 2020 onward were classified as occurring during the pandemic, as COVID-19 had already been introduced in Sweden by early 2020 and national public health recommendations were issued in March. Age-standardized incidence rates (ASR) per 1,000,000 children were calculated using the Segi 1960 World Standard Population [24], with 95% confidence intervals (CIs). Population data by age and year were obtained from Statistics Sweden [25]. Variance estimates were calculated using the formula proposed by Boniol and Heanue [26].

We assessed survival for first primary cancers diagnosed before or after January 1, 2020; this definition allowed children diagnosed in early 2020 to be analyzed within the treatment context shaped by the pandemic. Subsequent primary neoplasms were excluded to avoid immortal time bias. Unadjusted overall survival and corresponding 95% CIs at 3, 6, and 12 months post-diagnosis were estimated using the Kaplan–Meier estimator, both for all cancer types combined and by cancer type. We then employed logistic regression models to estimate adjusted odds ratios (aOR) and 95% CIs for mortality at 6 and 12 months after the childhood cancer diagnosis. Models were adjusted for age at diagnosis, sex, and maternal education—the latter being a well-established proxy for socioeconomic status known to influence childhood cancer outcomes [27]. Highest attained maternal education up to their child's cancer diagnosis was classified as low (primary

or lower secondary), medium (upper secondary), and high (postsecondary). A sensitivity analysis was performed using paternal education as an alternative proxy for socioeconomic status. Data on parental education were obtained from the Longitudinal Integrated Database for Health Insurance and Labor Market Studies (LISA) [28].

All analyses were performed in R version 4.4.2 [29]. The study protocol is registered on clinicaltrials.gov (NCT06482281). Two deviations from the registered protocol occurred: the 6-month survival analysis was added post hoc, and the originally planned inclusion of Danish data was not implemented due to data access constraints.

## Results

A total of 3,333 cancers were diagnosed to children and adolescents in Sweden between 2015 and 2022. Of these, 2,069 were diagnosed before the COVID-19 pandemic (2015–2019) and 1,264 during the pandemic years (2020–2022) (Table 1). The distribution of cancer types was broadly similar across the two periods, except for AML, which declined from 4.3% before 2020% to 2.5% from 2020 onwards. The most frequent cancer types were CNS tumors (26.6%), and leukemias (24.5%).

### Childhood cancer incidence before and during the COVID-19 pandemic

Fig 1 presents the absolute number of incident childhood cancer diagnoses per quarter, showing the 2015–2019 average alongside the individual years 2015–2022. Full quarterly counts for the 2015–2019 average and each pandemic year are provided in S1 Table, and a monthly version of the plot—using the aggregated 2015–2019 average for comparison—is available in S1 Fig. Overall, there was no major deviation during 2020 compared to the 2015–2019 average for any of the four quarters (Fig 1A).

In analyses by specific cancer type and diagnostic group, the number of ALL diagnoses declined slightly during the first two quarters of 2020 but subsequently increased in the second half of the year (Fig 1B). Notably, the decline in ALL diagnoses began as early as February 2020 (S1 Fig). Similarly, we observed a reduction in AML diagnoses during the first two quarters of 2020 compared to the 2015–2019 average, with additional declines in late 2021 and throughout 2022 (Fig 1C). However, the small absolute numbers are prone to random variation and limit interpretation. In contrast, we found a slight decrease in Hodgkin and non-Hodgkin lymphoma diagnoses in the third quarter of 2020 (Fig 1D and 1E) but not in the first quarter. For CNS tumor diagnoses, a marked increase was observed in 2020, particularly in the third quarter (Fig 1F).

Annual ASR per 1,000,000 children are presented in Table 2. When combining all childhood cancer diagnoses, no substantial changes in ASR were observed between the periods 2015–2019 and 2020–2022. However, the ASR of ALL decreased in 2020, followed by a pronounced increase in 2021, surpassing the 2015–2019 average. In contrast, the ASR of AML showed a marked decline throughout 2020–2022 compared to 2015–2019. For Hodgkin lymphomas, the ASR also decreased in 2020 and remained lower throughout 2020–2022 compared to 2015–2019. ASRs for non-Hodgkin lymphomas declined in 2020 but returned to pre-pandemic levels—and slightly exceeded them—in 2022. For CNS tumors, ASRs increased in 2020 but returned to 2015–2019 levels or below in 2021 and remained stable in 2022. ASRs for children younger than 15 years of age reveal similar results (S2 Table).

### Survival and mortality after a childhood cancer diagnosis before and during the COVID-19 pandemic

Estimates of 3-, 6-, and 12-month overall survival following childhood cancer diagnosis are presented in Table 3. The table provides cross-sectional summaries at each time point, indicating the number of individuals at risk of death at that time and the cumulative number of events that had occurred up to that time. Overall, 1-year survival was 94.8% (95% CI [93.9,95.8]) in the pre-pandemic period and 95.9% (95% CI [94.8,97.0]) during the pandemic, with no notable differences at earlier follow-up points. Patterns were similar across most diagnostic groups, with no pronounced changes in survival when comparing periods before and after the COVID-19 pandemic.

**Table 1. Characteristics of children aged 0–19 years newly diagnosed with cancer in Sweden, 2015–2022.**

|  | 2015–2019 | 2020–2022 | Overall |
|---|---|---|---|
|  | N = 2,069 | N = 1,264 | N = 3,333 |
| **Sex** |  |  |  |
| Male | 1,065 (51.5%) | 659 (52.1%) | 1,724 (51.7%) |
| Female | 1,004 (48.5%) | 605 (47.9%) | 1,609 (48.3%) |
| **Age (years)** |  |  |  |
| 0–4 | 613 (29.6%) | 395 (31.3%) | 1,008 (30.2%) |
| 5–9 | 384 (18.6%) | 209 (16.5%) | 593 (17.8%) |
| 10–14 | 422 (20.4%) | 241 (19.1%) | 663 (19.9%) |
| 15–19 | 650 (31.4%) | 419 (33.1%) | 1,069 (32.1%) |
| **Maternal education** |  |  |  |
| Low | 234 (11.3%) | 127 (10.0%) | 361 (10.8%) |
| Intermediate | 731 (35.3%) | 422 (33.4%) | 1,153 (34.6%) |
| High | 1,030 (49.8%) | 682 (54.0%) | 1,712 (51.4%) |
| Missing | 74 (3.6%) | 33 (2.6%) | 107 (3.2%) |
| **Paternal education** |  |  |  |
| Low | 255 (12.3%) | 151 (11.9%) | 406 (12.2%) |
| Intermediate | 899 (43.5%) | 521 (41.2%) | 1,420 (42.6%) |
| High | 779 (37.7%) | 496 (39.2%) | 1,275 (38.3%) |
| Missing | 136 (6.6%) | 96 (7.6%) | 232 (7.0%) |
| **Cancer type** |  |  |  |
| Leukemias | 528 (25.5%) | 288 (22.8%) | 816 (24.5%) |
| ALL | 391 (18.9%) | 240 (19.0%) | 631 (18.9%) |
| AML | 89 (4.3%) | 31 (2.5%) | 120 (3.6%) |
| Other Leukemias | 48 (2.3%) | 17 (1.3%) | 65 (2.0%) |
| Lymphomas | 243 (11.7%) | 148 (11.7%) | 391 (11.7%) |
| Hodgkin | 144 (7.0%) | 81 (6.4%) | 225 (6.8%) |
| Non-Hodkin | 72 (3.5%) | 46 (3.6%) | 118 (3.5%) |
| Other Lymphomas | 27 (1.3%) | 21 (1.7%) | 48 (1.4%) |
| CNS tumors | 547 (26.4%) | 341 (27.0%) | 888 (26.6%) |
| Ependymomas and choroid plexus tumor | 41 (2.0%) | 13 (1.0%) | 54 (1.6%) |
| Astrocytomas | 161 (7.8%) | 120 (9.5%) | 281 (8.4%) |
| High grade | 24 (1.2%) | 9 (0.7%) | 33 (1.0%) |
| Intracranial and intraspinal embryonal tumors | 76 (3.7%) | 42 (3.3%) | 118 (3.5%) |
| Other gliomas | 12 (0.6%) | 6 (0.5%) | 18 (0.5%) |
| Other specified intracranial and intraspinal neoplasms | 250 (12.1%) | 163 (12.9%) | 413 (12.4%) |
| Non-CNS solid tumors | 750 (36.2%) | 484 (38.3%) | 1,234 (37.0%) |

Abbreviations: ALL, acute lymphoblastic leukemia; AML, acute myeloid leukemia; CNS, central nervous system.

Adjusted mortality at 6 and 12 months following a childhood cancer diagnosis is presented in Fig 2. No increase in mortality was observed during the COVID-19 pandemic when combining all cancer types. Instead, there were indications, although not statistically significant, of reduced 1-year mortality among children diagnosed with ALL (aOR = 0.37; 95% CI [0.08,1.21]; $p$ = 0.14) and CNS tumors (aOR = 0.61, 95% CI [0.31,1.14]; $p$ = 0.13). These results were, however, based on only 5 and 15 deaths during 2020–2022, respectively. A similar pattern was observed for 6-month mortality; however, CIs were wide (Fig 2). An exploratory analysis of 3-month mortality also suggested lower mortality following a CNS tumor

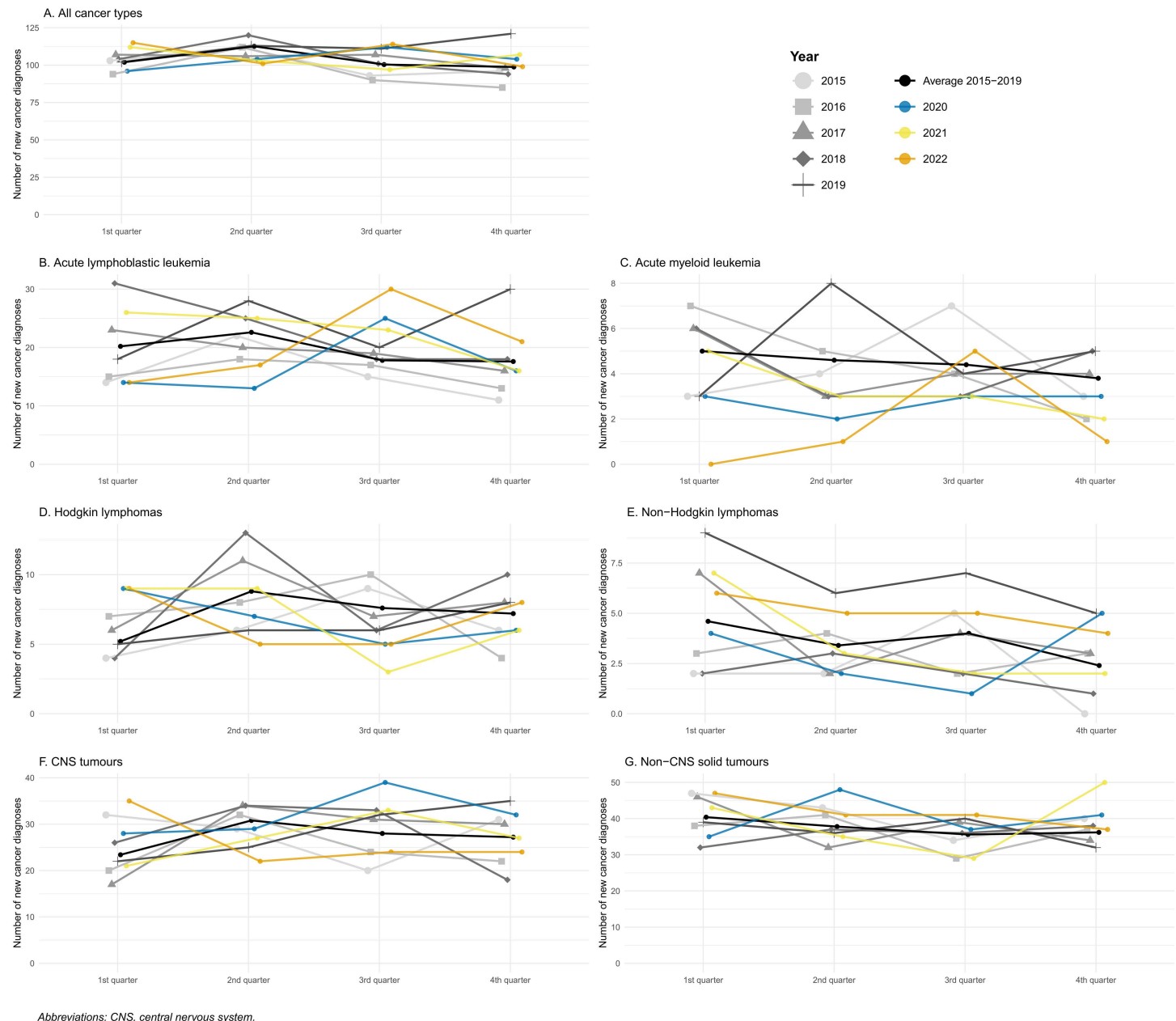

Abbreviations: CNS, central nervous system.

**Fig 1. Absolute number of new cancer diagnoses per quarter among children aged 0–19 years in Sweden, 2015–2022.**

diagnosis (aOR = 0.44, 95% CI [0.12,1.28]; $p$ = 0.16), although CIs were wide (S2 Fig). When exploring proportions of death by CNS tumor subtype, the most notable difference was seen for intracranial and intraspinal embryonal tumors: while 14.9% of children diagnosed with such a tumor died within one year from diagnosis in the pre-pandemic period, no such deaths were recorded during the pandemic period (S3 Table). Unadjusted ORs are shown in S3 Fig. Finally, two sensitivity analyses were performed, both yielding similar results. The first used paternal instead of maternal education as a proxy for parental socioeconomic status (S4 Fig), and the second included only children aged 0–14 years in the analysis (S5 Fig).

**Table 2. Age-standardized incidence rates (ASR) of cancer per 1,000,000 children aged 0–19 years in Sweden, 2015–2022.**

| | Age-standardized incidence rate (95% CI) | | | |
|---|---|---|---|---|
| | **2015–2019** | **2020** | **2021** | **2022** |
| **All cancer types** | 179.5 (172.9,186.1) | 174.7 (160.5,189.0) | 176.5 (161.6,191.5) | 181.2 (166.3,196.2) |
| ALL | 35.3 (31.9,38.7) | 30.1 (23.4,36.9) | 40.9 (32.7,49.2) | 37.5 (29.7,45.2) |
| AML | 8.0 (6.4,9.5) | 4.6 (2.4,6.7) | 5.4 (2.9,7.9) | 3 (1.1,4.8) |
| Hodgkin | 11.7 (10.5,12.9) | 10.4 (8.6,12.1) | 10.2 (8.1,12.3) | 10 (7.6,12.4) |
| Non-Hodgkin | 6.0 (4.9,7.0) | 4.6 (2.6,6.7) | 5.5 (3.1,7.8) | 7.8 (5.0,10.6) |
| CNS tumors | 47.0 (43.6,50.4) | 52.9 (45.0,60.7) | 44.9 (37.1,52.6) | 44.7 (36.8,52.6) |
| Non-CNS solid tumors | 65.1 (61.3,68.9) | 68.4 (59.6,77.3) | 65.4 (56.9,73.9) | 69.7 (60.9,78.4) |

**Abbreviations:** ALL, acute lymphoblastic leukemia; AML, acute myeloid leukemia; CNS, central nervous system; CI, confidence interval.

## Discussion

In this nationwide population-based study, we found no evidence of a decline in incident childhood cancer diagnoses in Sweden during the COVID-19 pandemic (2020–2022). The number of children diagnosed with ALL, however, declined in early 2020, followed by an increase in 2021; notably, the 2020 drop preceded the pandemic onset in Sweden. In contrast, AML diagnoses remained consistently lower throughout the pandemic. We observed a temporary decline in Hodgkin and non-Hodgkin lymphomas in the third quarter of 2020 and a transient increase in CNS tumors that normalized thereafter. Importantly, we found no evidence of increased 6-month or 1-year mortality following a childhood cancer diagnosis during the pandemic. Our findings might rather suggest a small decrease in mortality among children diagnosed with ALL and CNS tumors in 2020–2022 compared with the pre-pandemic period, although estimates were imprecise.

Our findings add to the sparse and heterogeneous evidence on the pandemic's impact on childhood cancer incidence. Contrary to our observation of no decline in overall childhood cancer incidence, a study from the United States reported a reduction in childhood cancer diagnoses overall, although its primary focus was adult cancers [4]. A more recent US analysis reported a 5% decrease in childhood cancer incidence in 2020 compared with 2019, with incidence returning to baseline by late 2020; unlike our findings, no decline in leukemias was observed [8]. In Italy, pediatric solid tumors diagnoses dropped early in the pandemic [9]. Although we also observed a transient decline in lymphoma diagnoses, our data showed increases in CNS and non-CNS solid tumor diagnoses in 2020—the latter remaining elevated in subsequent years. Consistent with our findings, a marked decline in ALL diagnoses early in the pandemic was observed in data from the south-eastern region of Norway [11]. In contrast, studies from Greece, France, and Germany found no evidence of a decline in childhood cancer diagnoses [12–15]. Notably, the German study reported increases across all diagnostic groups in 2020, which the authors hypothesized could, at least for some solid tumors, be due to potential heightened parental vigilance and earlier detection [14,15].

Variations between studies likely reflect differences in study design, case definitions, and follow-up time. Most earlier studies covered only the first months of the pandemic, restricting their ability to capture longer-term patterns or rebounds. In contrast, our three-year study observed both initial disruptions and subsequent shifts in incidence. Sweden's healthcare infrastructure, welfare system, pandemic course, and related measures also likely influenced our results. Unlike most other European countries, Sweden avoided strict lockdowns; this allows for examining how childhood cancer incidence and outcomes evolved in a setting with comparatively fewer disruptions to daily life and healthcare access. Despite this lack of prolonged lockdowns, indirect health system effects of the pandemic—such as temporary strain on diagnostic and treatment services—cannot be precluded and have been reported globally in settings with both high and low COVID-19 burden [30].

While we estimated that overall cancer incidence remained stable during the COVID-19 pandemic, patterns by diagnostic group may reflect diagnostic disruptions or shifts in disease triggers. The decline in ALL cases in early 2020,

**Table 3. Overall survival (OS) estimates at 3 months, 6 months, and 1 year after cancer diagnosis among children aged 0–19 years in Sweden, comparing the pre-pandemic period (2015–2019) to the pandemic period (2020–2022).**

| | 2015-2019 | | | 2020-2022 | | |
|---|---|---|---|---|---|---|
| | Number of children at risk[a] | Cumulative number of deaths[b] | OS (95% CI) | Number of children at risk[a] | Cumulative number of deaths[b] | OS (95% CI) |
| **All cancer types** | | | | | | |
| Start | 1,991 | | | 1,232 | | |
| 3-month | 1,961 | 30 | 98.5 (98.0,99.0) | 1,215 | 17 | 98.6 (98.0,99.3) |
| 6-month | 1,938 | 53 | 97.3 (96.6,98.0) | 1,204 | 28 | 97.7 (96.9,98.6) |
| 1-year | 1,888 | 103 | 94.8 (93.9,95.8) | 1,182 | 50 | 95.9 (94.8,97.0) |
| **ALL** | | | | | | |
| Start | 384 | | | 239 | | |
| 3-month | 381 | 3 | 99.2 (98.3,100.0) | 234 | 5 | 97.9 (96.1,99.7) |
| 6-month | 377 | 7 | 98.2 (96.8,99.5) | 234 | 5 | 97.9 (96.1,99.7) |
| 1-year | 370 | 14 | 96.4 (94.5,98.2) | 234 | 5 | 97.9 (96.1,99.7) |
| **AML** | | | | | | |
| Start | 84 | | | 28 | | |
| 3-month | 77 | 7 | 91.7 (85.9,97.8) | 27 | 1 | 96.4 (89.8,100.0) |
| 6-month | 77 | 7 | 91.7 (85.9,97.8) | 25 | 3 | 89.3 (78.5,100.0) |
| 1-year | 73 | 11 | 86.9 (80.0,94.4) | 24 | 4 | 85.7 (73.7,99.7) |
| **Hodgkin** | | | | | | |
| Start | 144 | | | 80 | | |
| 3-month | 144 | 0 | 100.0 (100.0,100.0) | 80 | 0 | 100.0 (100.0,100.0) |
| 6-month | 144 | 0 | 100.0 (100.0,100.0) | 80 | 0 | 100.0 (100.0,100.0) |
| 1-year | 144 | 0 | 100.0 (100.0,100.0) | 80 | 0 | 100.0 (100.0,100.0) |
| **Non-Hodkin** | | | | | | |
| Start | 70 | | | 46 | | |
| 3-month | 68 | 2 | 97.1 (93.3,100.0) | 45 | 1 | 97.8 (93.7,100.0) |
| 6-month | 64 | 6 | 91.4 (85.1,98.2) | 44 | 2 | 95.7 (89.9,100.0) |
| 1-year | 62 | 8 | 88.6 (81.4,96.3) | 44 | 2 | 95.7 (89.9,100.0) |
| **CNS tumors** | | | | | | |
| Start | 533 | | | 338 | | |
| 3-month | 520 | 13 | 97.6 (96.3,98.9) | 334 | 4 | 98.8 (97.7,100.0) |
| 6-month | 514 | 19 | 96.4 (94.9,98.0) | 329 | 9 | 97.3 (95.6,99.1) |
| 1-year | 499 | 34 | 93.6 (91.6,95.7) | 323 | 15 | 95.6 (93.4,97.8) |
| **Non-CNS solid tumors** | | | | | | |
| Start | 701 | | | 463 | | |
| 3-month | 696 | 5 | 99.3 (98.7,99.9) | 459 | 4 | 99.1 (98.3,100.0) |
| 6-month | 688 | 13 | 98.1 (97.2,99.1) | 456 | 7 | 98.5 (97.4,99.6) |
| 1-year | 669 | 32 | 95.4 (93.9,97.0) | 442 | 21 | 95.5 (93.6,97.4) |

[a] Refers to the number of children alive and therefore still at risk of death at each follow-up time point.

[b] Represents all deaths that had occurred up to that time point.

**Abbreviations:** ALL, acute lymphoblastic leukemia; AML, acute myeloid leukemia; CNS, central nervous system.

followed by a rebound in 2021, could suggest delays in diagnoses. However, this interpretation is complicated by two factors. First, ALL typically presents acutely, leaving little room for delayed diagnosis. Second, the timing of the decline—which began in February—preceded any substantial impact of the pandemic on Swedish society or healthcare services.

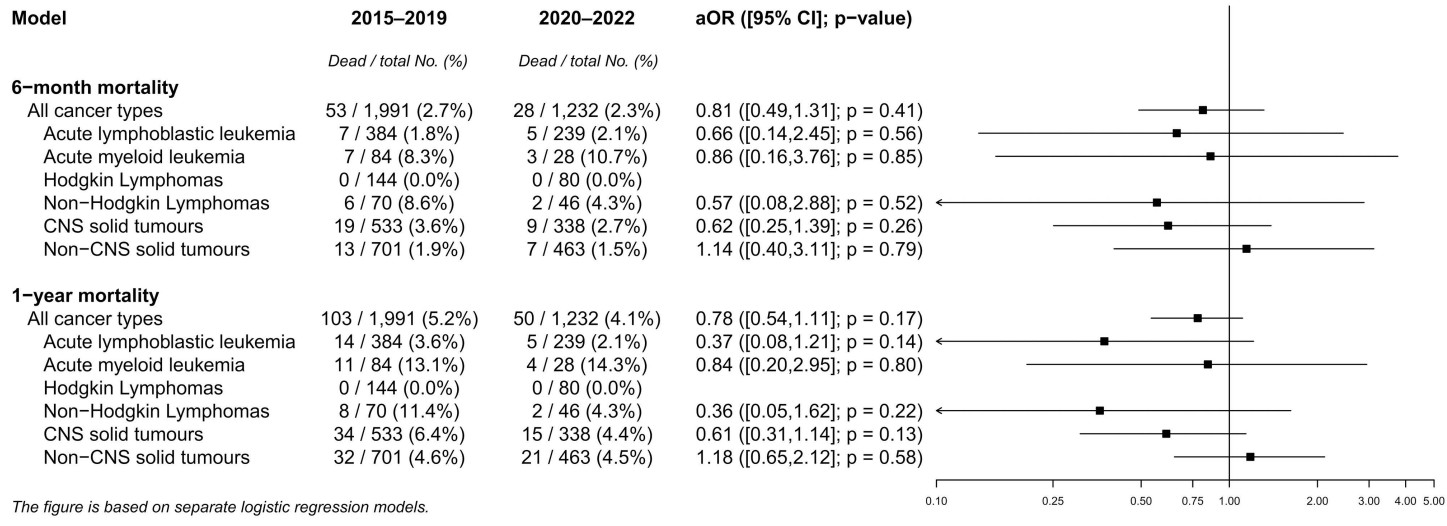

| Model | 2015–2019 | 2020–2022 | aOR ([95% CI]; p−value) | |
|---|---|---|---|---|
| | *Dead / total No. (%)* | *Dead / total No. (%)* | | |
| **6−month mortality** | | | | |
| All cancer types | 53 / 1,991 (2.7%) | 28 / 1,232 (2.3%) | 0.81 ([0.49,1.31]; p = 0.41) | |
| Acute lymphoblastic leukemia | 7 / 384 (1.8%) | 5 / 239 (2.1%) | 0.66 ([0.14,2.45]; p = 0.56) | |
| Acute myeloid leukemia | 7 / 84 (8.3%) | 3 / 28 (10.7%) | 0.86 ([0.16,3.76]; p = 0.85) | |
| Hodgkin Lymphomas | 0 / 144 (0.0%) | 0 / 80 (0.0%) | | |
| Non−Hodgkin Lymphomas | 6 / 70 (8.6%) | 2 / 46 (4.3%) | 0.57 ([0.08,2.88]; p = 0.52) | |
| CNS solid tumours | 19 / 533 (3.6%) | 9 / 338 (2.7%) | 0.62 ([0.25,1.39]; p = 0.26) | |
| Non−CNS solid tumours | 13 / 701 (1.9%) | 7 / 463 (1.5%) | 1.14 ([0.40,3.11]; p = 0.79) | |
| **1−year mortality** | | | | |
| All cancer types | 103 / 1,991 (5.2%) | 50 / 1,232 (4.1%) | 0.78 ([0.54,1.11]; p = 0.17) | |
| Acute lymphoblastic leukemia | 14 / 384 (3.6%) | 5 / 239 (2.1%) | 0.37 ([0.08,1.21]; p = 0.14) | |
| Acute myeloid leukemia | 11 / 84 (13.1%) | 4 / 28 (14.3%) | 0.84 ([0.20,2.95]; p = 0.80) | |
| Hodgkin Lymphomas | 0 / 144 (0.0%) | 0 / 80 (0.0%) | | |
| Non−Hodgkin Lymphomas | 8 / 70 (11.4%) | 2 / 46 (4.3%) | 0.36 ([0.05,1.62]; p = 0.22) | |
| CNS solid tumours | 34 / 533 (6.4%) | 15 / 338 (4.4%) | 0.61 ([0.31,1.14]; p = 0.13) | |
| Non−CNS solid tumours | 32 / 701 (4.6%) | 21 / 463 (4.5%) | 1.18 ([0.65,2.12]; p = 0.58) | |

*The figure is based on separate logistic regression models.*

*All models are adjusted by age, sex, and maternal education.*

*Abbreviations: CNS, central nervous system; aOR, adjusted Odds Ratio; CI, Confidence Interval; p, p−value.*

**Fig 2. Adjusted odds ratios (aOR) of 6-month and 1-year mortality after cancer diagnosis among children aged 0–19 years in Sweden, comparing the pandemic period (2020–2022) to the pre-pandemic period (2015–2019).**

This timing is also noteworthy in light of renewed discussion, prompted by the COVID-19 pandemic, about whether substantially altered infection exposure could transiently influence ALL incidence [31–33]. According to the "delayed infection hypothesis," exposure to common infections during infancy may train the immune system and protect against ALL during later childhood, whereas delayed exposure may instead trigger the disease in genetically susceptible children [16]; however, the underlying biological mechanisms remain uncertain. Under this framework, a period of reduced transmission of common infections—whether due to official restrictions or informal behavioral changes such as improved hygiene and voluntary social distancing—has been proposed as one possible explanation for a short-term decline in incidence through reduced exposure to potential immunologic triggers. Such a decline was observed during the 2003 SARS outbreak in Hong Kong [17]. Notably, even before formal restrictions were implemented in Sweden, voluntary behavioral changes may have already influenced infection patterns. Still, it is unlikely that these changes explain the observed decline in ALL diagnoses as early as February. Therefore, our data do not support this explanation, and random variation remains the most plausible interpretation of the ALL patterns observed.

For the remaining diagnostic groups, distinct patterns were observed that may reflect different underlying mechanisms. The persistent reduction in our estimates of AML incidence throughout 2020–2022 lacks a clear explanation and may reflect random variation. Lymphoma diagnoses declined briefly in mid-2020, consistent with temporary diagnostic delays. However, only non-Hodgkin lymphomas had returned to or exceeded pre-pandemic levels by the end of the study. As non-Hodgkin lymphomas are generally less prone to diagnostic delays due to their acute presentation, such delays may be less likely to explain the observed fluctuations. CNS tumors showed a transient spike in 2020, particularly in the third quarter, which lacks a definitive explanation. This increase may represent random variation or possibly earlier detection due to increased parental or physician vigilance during the early pandemic period [14].

Despite these incidence fluctuations, we observed no corresponding increase in mortality up to one year after diagnosis. This is notable, as delays and disruptions in diagnosis or treatment are expected to lead to more advanced disease and worse outcomes [18]. This apparent stability—or in some cases, indication of improvement—in survival could reflect several factors. Reduced circulation of common infections during the pandemic may have lowered the risk of

treatment-related complications, particularly in children undergoing immunosuppressive therapy. Conversely, one might expect SARS-CoV-2 infections to worsen outcomes, but despite high community transmission in Sweden, we found no indication of poorer survival among children with cancer. This may reflect the generally mild course of COVID-19 in pediatric populations [34], or the resilience of pediatric oncology care under strained healthcare conditions. The observed tendency of improvement in survival for ALL may also partly reflect the introduction of the European ALLTogether treatment protocol [35], which started in Sweden in August 2019. As this protocol was introduced shortly before the pandemic, we cannot disentangle its potential survival benefits from pandemic-related effects in our analysis.

Key strengths of our study include its nationwide scope, high-quality registry data, and near-complete follow-up. By examining both incidence and mortality, the study provides a comprehensive assessment of the pandemic's potential impact on childhood cancer care. The inclusion of parental education as a proxy for socioeconomic status also allowed us to account for potential social gradients in access to care. Our study period extended through the end of 2022, offering longer follow-up than most previous studies and capturing not only early disruptions but also longer-term patterns.

However, some limitations should be acknowledged. First, due to the low number of cases in some diagnostic groups, our ability to draw firm conclusions is constrained by random variation and wide CIs, reflecting substantial statistical uncertainty around several estimates. Moreover, although registry data provide robust incidence and mortality estimates, they lack detailed clinical information, such as stage at diagnosis, limiting insights into subtler childhood cancer care disruptions. Therefore, the lack of increased mortality does not rule out other consequences of disrupted care (e.g., more intensive treatment, higher relapse rates), which have been reported globally during the pandemic [30]. Similarly, without information on diagnostic route or symptom onset, we cannot directly assess diagnostic delays—only infer from patterns in our incidence estimates.

Our findings suggest that the Swedish pediatric oncology system remained largely resilient during the COVID-19 pandemic, as no major changes were observed on overall childhood cancer incidence or survival up to one-year post-diagnosis. While patterns of incidence estimates varied by cancer type, the timing of some changes and acute presentation of certain pediatric cancers, such as ALL and non-Hodgkin lymphoma, argue against widespread diagnostic delay. Moreover, the absence of increased mortality suggests that, even if present, any delays or treatment disruptions were limited or successfully mitigated; results instead suggested a small decline in mortality for ALL and CNS tumors. These findings underscore the importance of robust, adaptable healthcare systems capable of maintaining essential services during global crises. However, the results reflect the Swedish context and may not be generalizable to settings with different pandemic course, responses, or healthcare infrastructure. Future research should assess longer-term incidence and survival patterns in the post-pandemic period, particularly as children resume pre-pandemic patterns of social interaction. This may be particularly relevant for ALL, given hypotheses linking infection exposure in infancy to its pathogenesis. Finally, incorporating data on disease stage at diagnosis and treatment trajectories may offer a more comprehensive understanding of the pandemic's impact on childhood cancer care.

## Supporting information

**S1 Table. Absolute number of new cancer diagnoses per quarter among children aged 0–19 years in Sweden, 2015–2022.**
(PDF)

**S2 Table. Age-standardized incidence rates (ASR) of cancer per 1,000,000 children aged 0–14 years in Sweden, 2015–2022.**
(PDF)

**S3 Table. Proportion of deaths within 6 months and 1 year after a CNS tumor diagnosis among children aged 0–19 years in Sweden, 2015–2022.**
(PDF)

**S1 Fig. Absolute number of new cancer diagnoses per month among children aged 0–19 years in Sweden, 2015–2022.**
(PDF)

**S2 Fig. Adjusted odds ratios (aOR) of 3-month mortality after cancer diagnosis among children aged 0–19 years in Sweden, comparing the pandemic period (2020–2022) to the pre-pandemic period (2015–2019).**
(PDF)

**S3 Fig. Unadjusted odds ratios (aOR) of 6-month and 1-year mortality after cancer diagnosis among children aged 0–19 years in Sweden, comparing the pandemic period (2020–2022) to the pre-pandemic period (2015–2019).**
(PDF)

**S4 Fig. Adjusted odds ratios (aOR) of 6-month and 1-year mortality after cancer diagnosis among children aged 0–19 years in Sweden, comparing the pandemic period (2020–2022) to the pre-pandemic period (2015–2019), adjusted by paternal education.**
(PDF)

**S5 Fig. Adjusted odds ratios (aOR) of 6-month and 1-year mortality after cancer diagnosis among children aged 0–14 years in Sweden, comparing the pandemic period (2020–2022) to the pre-pandemic period (2015–2019).**
(PDF)

**S1 Checklist. Reporting of Studies Conducted using Observational Routinely-Collected Data (RECORD) guideline checklist.**
(PDF)

## Author contributions

**Conceptualization:** Christina-Evmorfia Kampitsi, Javier Louro, Maria Feychting, Giorgio Tettamanti.

**Data curation:** Javier Louro, Giorgio Tettamanti.

**Formal analysis:** Javier Louro.

**Funding acquisition:** Giorgio Tettamanti.

**Methodology:** Christina-Evmorfia Kampitsi, Javier Louro, Hanna Mogensen, Friederike Erdmann, Kleopatra Georgantzi, Mats Heyman, Päivi Lähteenmäki, Anna Nilsson, Maria Feychting, Giorgio Tettamanti.

**Project administration:** Maria Feychting, Giorgio Tettamanti.

**Supervision:** Maria Feychting, Giorgio Tettamanti.

**Visualization:** Christina-Evmorfia Kampitsi, Javier Louro, Maria Feychting, Giorgio Tettamanti.

**Writing – original draft:** Christina-Evmorfia Kampitsi, Javier Louro.

**Writing – review & editing:** Christina-Evmorfia Kampitsi, Javier Louro, Hanna Mogensen, Friederike Erdmann, Kleopatra Georgantzi, Mats Heyman, Päivi Lähteenmäki, Anna Nilsson, Maria Feychting, Giorgio Tettamanti.

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
