## [Editor Report · Decision Letter 0]

6 Oct 2025

Dear Dr Kampitsi,

Thank you for submitting your manuscript entitled "The impact of the COVID-19 pandemic on childhood cancer in Sweden: temporal patterns in incidence and survival" for consideration by PLOS Medicine.

Your manuscript has now been evaluated by the PLOS Medicine editorial staff as well as by an academic editor with relevant expertise and I am writing to let you know that we would like to send your submission out for external peer review.

For clinical studies, please upload a copy of your trial study protocol as a supporting information file. The study protocol should be the version submitted for approval to the institutional review board or ethics committee, should include any amendments to the study protocol, as well as the date of their approval by the institutional review or ethics committee. Please also detail any deviations from the study protocol in the Methods section of your manuscript. The editors will consider the protocol and study conduct prior to a final decision for external review.

Please re-submit your manuscript within two working days, i.e. by Oct 08 2025 11:59PM.

Kind regards,

Heather Van Epps, PhD

Consulting Editor

PLOS Medicine

---

## [Decision Letter · Decision Letter 1]

12 Nov 2025

Dear Dr Kampitsi,

Many thanks for submitting your manuscript "The impact of the COVID-19 pandemic on childhood cancer in Sweden: temporal patterns in incidence and survival" (PMEDICINE-D-25-03481R1) to PLOS Medicine. The paper has been reviewed by subject experts and a statistician; their comments are included below and can also be accessed here: [LINK]

As you will see, the reviewers provided mixed feedback, offering suggestions as well as critical comments. After discussing the paper with the editorial team and an academic editor with relevant expertise, I'm pleased to invite you to revise the paper in response to the reviewers' comments. We plan to send the revised paper to some or all of the original reviewers, and we cannot provide any guarantees at this stage regarding publication.

We ask that you submit your revision by Dec 03 2025. However, if this deadline is not feasible, please contact me by email, and we can discuss a suitable alternative.

Don't hesitate to contact me directly with any questions (atosun@plos.org).

Best regards,

Alexandra

Alexandra Tosun, PhD

Senior Editor

PLOS Medicine

atosun@plos.org

Comments from the reviewers:

Reviewer #1: The authors wanted to assess the possible impact of the COVID-19 epidemic on incidence and survival of childhood cancer in Sweden. Data from pre-pandemic years 2015-2019 were compared with those from the pandemic years 2020-2022.

Although the topic is important, I am not impressed by the paper. Comments are listed below:

Introduction:

Line 17, ref. 11: Data in ref. 11 are from the Oslo University Hospital. The authors should have checked NORDCAN data from all of Norway, where this tendency is not seen.

Methods:

Line 72: It is right to exclude "subsequent primary neoplasms" in the survival analysis. However, the authors should have provided more absolute numbers. In Table 1, there are 1987 cancers in 2015-1019 and 1230 in 2020-2022. In Table 3, at 3-months there are 1957 cancers at risk in 2015-2019 and 1120 in 2020-2022. This means that in 2015-2019, 98% (1957/1987) of the cancers were first primary cancers, while in 2020-2022 this was 91% (1120/1230). The authors should have commented on this difference. Or maybe I misread the table, but it should be made clear what the column "at risk" means?

Results:

Line 111: It is started correctly that … "no significant changes…". But a similar reservation is not used later.

Table 3: What does the column "at risk" refer to? At risk at what time?

Figure 2: Now the total number of cases is again 1987 for 2015-2019 and now 1229 for 2020-2022. So does this mean that there was no subsequent primary in 2015-2019 and one in 2020-2022?

Discussion.

Lines 147-157: It should be made clear that none of these changes are statistically significant.

Lines 167-168: Well check NORDCAN. These differences may reflect changes in referral patterns rather than changes in incidence. And not statistically significant.

Lines 188-198: I do not follow the discussion here, as the authors started explicitly, that the decline in ALL started in February 2020, before the pandemic had any substantial impact on the Swedish society.

Lines 199-208: repetition of Results, can be deleted.

Lines 237-239: Very important, should come first, as this is a major limitation of the study.

Line 250: The authors state the interest in assessing long-term trends in incidence and survival. So why did they end the data collection with 2022?

Abstract.

Para starting with "overall incidence …": The data did not prove a decline. Say instead, that a tendency was seen…

Same para: The last sentence can go out as these results build on so few cases that it is not worth reporting in an abstract.

Reviewer #2: Kampitsi et al present a well-executed register-based analysis of childhood cancer incidence and short-term survival in Sweden during the COVID-19 pandemic. The authors leverage comprehensive national registry data covering 2015 to 2022 to examine whether the pandemic influenced diagnostic rates or outcomes. The study addresses an important question given inconsistent published data. Overall, the paper is of valuable, although requires some minor edits prior to publication.

* While the study cautiously interprets observed changes in specific cancer types (e.g., the early decline and rebound in ALL), the language in parts of the Discussion may overreach. The circulation of common infections and the relationship with the development of pediatric cancers (specifically ALL) is a theory of association, without clear causal linkage.

* To the prior point, although it makes sense to divide the data as pre- and during the pandemic it may oversimplify the data and negate some of the inherent natural fluctuations over time. Segregation of data from year to year in some of the presented data especially between 2015 and 2022, for example cancer disease types, would add context to some of the claims of pediatric cancer distribution changes during the pandemic.

* Although there may have not been prolonged lockdown, health system deficiencies secondary to the pandemic (saturation of services, etc) are likely still are play. The largest study on the possible "indirect" impact of COVID-19 on pediatric cancer services is Graetz D, et al. Lancet Child Adolesc Health. 2021. That would be a valuable reference to cite and contextualize the data from Sweden.

Reviewer #3: This manuscript presents an examination of childhood cancer rates in Sweden during the years prior to and during the COVID-19 pandemic, using registry data from national Swedish registries. My comments are focussed on the statistical analysis, which seems to be largely appropriate. Further, the conclusions drawn appear to be supported by the data. However, I do have some suggestions and comments for improvements.

1. What dates were used to define the pandemic? It seems that years are used to define pandemic and non-pandemic time periods, but given that a pandemic was only declared in March of 2020, would it be more appropriate to consider the first quarter of 2020 as part of the non-pandemic period?

2. I found Figure 1 (and Supplementary Figure 1) somewhat difficult to interpret. I think it would be easier to interpret a figure with four lines on it: one for each year 2020-2022, and one for average 2015-2019. The y-axis would be the absolute number of new cancer diagnoses; the x-axis would be 1st quarter, 2nd quarter, 3rd quarter, and 4th quarter.

3. I have two further comments on Figure 1 (which also apply to Supplementary Table 1, 2, Supplementary Figure 1). First, why is it appropriate to group together the years 2015-2019? I think it would be more appropriate to consider each of the years 2015-2019 separately, to get some idea of the changes over those years and the variability to be expected from year to year. Second, why are the years split into quarters? Why not months? Why is it necessary to split years up at all? Is there a seasonal component to cancer diagnoses that is captured through a split into quarters?

---

* Please upload any figures associated with your paper as individual TIF or EPS files with 300dpi resolution at resubmission; please read our figure guidelines for more information on our requirements: http://journals.plos.org/plosmedicine/s/figures. While revising your submission, we strongly recommend that you use PLOS's NAAS tool (https://ngplosjournals.pagemajik.ai/artanalysis) to test your figure files. NAAS can convert your figure files to the TIFF file type and meet basic requirements (such as print size, resolution), or provide you with a report on issues that do not meet our requirements and that NAAS cannot fix.

After uploading your figures to PLOS's NAAS tool - https://ngplosjournals.pagemajik.ai/artanalysis, NAAS will process the files provided and display the results in the "Uploaded Files" section of the page as the processing is complete.

If the uploaded figures meet our requirements (or NAAS is able to fix the files to meet our requirements), the figure will be marked as "fixed" above. If NAAS is unable to fix the files, a red "failed" label will appear above.

When NAAS has confirmed that the figure files meet our requirements, please download the file via the download option, and include these NAAS processed figure files when submitting your revised manuscript.

FIGURES AND TABLES

SUPPLEMENTARY MATERIAL

REFERENCES

STUDY TYPE-SPECIFIC REQUESTS

* Abstract: Please include the study design, population and setting, number of participants, years during which the study took place (enrollment and follow up), length of follow up, and main outcome measures.

* Please ensure that the study is reported according to the RECORD guideline (available from https://www.record-statement.org) and include the completed checklist as Supporting Information. Please add the following statement, or similar, to the Methods: "This study is reported as per the Reporting of Studies Conducted using Observational Routinely-Collected Data (RECORD) guideline (S1 Checklist)." When completing the checklist, please use section and paragraph numbers, rather than page numbers.

* For all observational studies, in the manuscript text, please indicate: (1) the specific hypotheses you intended to test, (2) the analytical methods by which you planned to test them, (3) the analyses you actually performed, and (4) when reported analyses differ from those that were planned, transparent explanations for differences that affect the reliability of the study's results. If a reported analysis was performed based on an interesting but unanticipated pattern in the data, please be clear that the analysis was data driven.

* Please state in the Methods section whether the study had a prospective protocol or analysis plan. If a prospective analysis plan (from your funding proposal, IRB or other ethics committee submission, study protocol, or other planning document written before analyzing the data) was used in designing the study, please include the relevant document(s) with your revised manuscript as a Supporting Information file to be published alongside your study and cite it in the Methods section. A legend for this file should be included at the end of your manuscript. If no such document exists, please make sure that the Methods section transparently describes when analyses were planned, and when/why any data-driven changes to analyses took place. Changes in the analysis, including those made in response to peer review comments, should be identified as such in the Methods section of the paper, with rationale.

---

## [Decision Letter · Decision Letter 2]

16 Jan 2026

Dear Dr. Kampitsi,

Thank you very much for re-submitting your manuscript "The impact of the COVID-19 pandemic on childhood cancer in Sweden: temporal patterns in incidence and survival" (PMEDICINE-D-25-03481R2) for review by PLOS Medicine.

Thank you for your detailed response to the reviewers' and editors’ comments. I have discussed the paper with my colleagues and the academic editor, and it has also been seen again by all of the original reviewers. The changes made to the paper were mostly satisfactory to the reviewers. As such, we intend to accept the paper for publication, pending your attention to the reviewers' and editors' comments below in a further revision. When submitting your revised paper, please once again include a detailed point-by-point response to the editorial comments. The remaining issues that need to be addressed are listed at the end of this email.

In revising the manuscript for further consideration here, please ensure you address the specific points made by each reviewer and the editors. In your rebuttal letter you should indicate your response to the reviewers' and editors' comments and the changes you have made in the manuscript. Please submit a clean version of the paper as the main article file. A version with changes marked must also be uploaded as a marked up manuscript file. Please also check the guidelines for revised papers at http://journals.plos.org/plosmedicine/s/revising-your-manuscript for any that apply to your paper.

We ask that you submit your revision by Jan 23 2026. However, if this deadline is not feasible, please contact me (atosun@plos.org) or the journal staff by email, and we can discuss a suitable alternative.

We look forward to receiving the revised manuscript.

Sincerely,

Alexandra Tosun, PhD

Senior Editor

PLOS Medicine

plosmedicine.org

Comments from Reviewers:

Reviewer #1: none

Reviewer #2: The authors have comprehensively addressed the comments from the first submission.

Reviewer #3: I thank the authors for their responses to my comments.

My view is that Figure 1 in the paper should be replaced by a variant of Figure S2, which I believe displays the salient information about variability between years more clearly than Figure 1. Suggestions for further improvements to this figure are to use separate plotting symbols for each year. It may also be helpful to colour the 2015-2019 years in various shades of grey and overlay the average 2015-2019 counts using a thicker black line.

Requests from Editors:

GENERAL

* Please confirm that your title complies with to PLOS Medicine's style. Your title must be nondeclarative and not a question. It should begin with main concept if possible. "Effect of" (or “Impact of”) should be used only if causality can be inferred, i.e., for an RCT. Please place the study design ("A randomized controlled trial," "A retrospective study," "A modelling study," etc.) in the subtitle (ie, after a colon).

* Statistical reporting: Please revise throughout the manuscript, including tables and figures.

- Please report statistical information as follows to improve clarity for the reader, " "XX% (95% CI [XX,YY]; p</=)" ".

- Please separate upper and lower bounds with commas instead of hyphens as the latter can be confused with reporting of negative values.

- Please repeat statistical definitions (HR, CI etc.) for each set of parentheses.

* Please ensure that all abbreviations are defined at first use throughout the text (including statistical abbreviations).

* Please ensure that tables and figures, including those in supplementary files, are appropriately referenced in the main text. We noted that, starting with Figure S2/Figure S3, the referencing is incorrect.

* Please review your text for claims of novelty or primacy (e.g. 'for the first time' or ‘novel’) and remove this language.

* Please confirm that any use of statistical terms (such as trend or significant) are supported by the data, and if not please remove them. The term trend should be used only when the test for trend has been conducted.

* Please define all acronyms used in each figure or table in the corresponding legend.

* Please confirm that you used patient-centered language. Please note that patient-centered language is constructed with the use of post-modified nouns putting the person first in the sentence structure.

* Please review your manuscript and edit to ensure compliance with our inclusive language requirements (https://journals.plos.org/plosmedicine/s/human-subjects-research#loc-categorization).

* Please consider including an acknowledgment of study participants and of individuals who played a role in data collection or participant care.

* Your study is observational and therefore causality cannot be inferred. Please remove language that implies causality and refer to associations instead (e.g. Abstract Conclusions or Author Summary).

ABSTRACT

* Please confirm that your abstract complies with our requirements, including providing all the information relevant to this study type https://journals.plos.org/plosmedicine/s/submission-guidelines#loc-abstract

* Please confirm that all numbers presented in the abstract are present and identical to numbers presented in the main manuscript text.

* Please quantify the results on incidence rates (with 95% CIs and p values).

* Abstract Conclusions: Please revise with regard to using associational language.

* Please specify which cancer types were included/excluded (e.g. “we included non-malignant central nervous system (CNS) tumors and intracranial/intraspinal germ cell tumors” as done in your Methods section).

* In the abstract, please include the important dependent variables that are adjusted for in the analyses.

METHODS AND RESULTS

* Please remove ‘Role of the funding source’ from the Methods section. This information should only be included in the metadata in the online submission form.

* When discussing age, please add ‘years’ as a unit (including tables and figures).

* Figures: Please ensure to provide a label for the y-axis.

* Figure 1: We agree with Reviewer #3 that the visualization of the information in Figure 1 could be improved, and we strongly recommend following their advice.

* Table 2: For CNS tumors, you describe the difference between the 2015–2019 levels (47.0 (43.6–50.4)) and the 2020 levels (44.9 (37.1–52.6)) as comparable (a difference of 2.1). However, for Hodgkin, you describe a difference of 1.3 (with overlapping confidence interval (CI) values) as a decrease. Please explain how you determined whether there was a decrease, increase, or stable pattern.

* “Instead, there were indications of reduced 1-year mortality…” – based on the confidence intervals and the small n-numbers, we believe this is an overstatement. Please revise.

* An exploratory analysis of 3-month mortality also suggested lower mortality following a CNS tumor diagnosis (aOR = 0.44, 95% CI [0.12–1.28]; p = 0.161) (S2 Figure). – please reference S3 Figure here instead and see our comment above.

* Please confirm that where relevant figures include 95% CIs.

* Please confirm you provided the unadjusted comparisons as well as the adjusted comparisons in all relevant Tables.

* Please confirm you specified the variables controlled for in all relevant Tables.

DISCUSSION

* Due to the small number of cases and wide confidence intervals, we recommend carefully wording your findings and their interpretation (e.g., "Our findings rather suggest lower mortality...").

* Please remove the 'conclusions' subheading from the discussion. Please also remove any other subheadings from the discussion.

General Editorial Requests

---

## [Editor Report · Decision Letter 3]

27 Jan 2026

Dear Dr Kampitsi,

On behalf of my colleagues and the Academic Editor, Lars Åke Persson, I am pleased to inform you that we have agreed to publish your manuscript "Childhood cancer in Sweden during the COVID-19 pandemic: Temporal patterns in incidence and survival in a nationwide register-based cohort study" (PMEDICINE-D-25-03481R3) in PLOS Medicine.

I appreciate your thorough responses to the reviewers' and editors' comments throughout the editorial process. We look forward to publishing your manuscript, and editorially there is only one remaining point that should be addressed prior to publication. We will carefully check whether the change has been made. If you have any questions or concerns regarding these final requests, please feel free to contact me at atosun@plos.org.

Please see below the minor point that we request you respond to:

*Table 1: Please add 'years' as unit for "Age".

Before your manuscript can be formally accepted you will need to complete some formatting changes, which you will receive in a follow up email (including the editorial request above). Please be aware that it may take several days for you to receive this email; during this time no action is required by you. Once you have received these formatting requests, please note that your manuscript will not be scheduled for publication until you have made the required changes.

PRESS

Sincerely,

Alexandra Tosun, PhD

Senior Editor

PLOS Medicine